# Preparation and Activity of Hemostatic and Antibacterial Dressings with Greige Cotton/Zeolite Formularies Having Silver and Ascorbic Acid Finishes

**DOI:** 10.3390/ijms242317115

**Published:** 2023-12-04

**Authors:** J. Vincent Edwards, Nicolette T. Prevost, Doug J. Hinchliffe, Sunghyun Nam, SeChin Chang, Rebecca J. Hron, Crista A. Madison, Jade N. Smith, Chelsie N. Poffenberger, Michelle M. Taylor, Erika J. Martin, Kirsty J. Dixon

**Affiliations:** 1Southern Regional Research Center, Agricultural Research Service, United States Department of Agriculture, New Orleans, LA 70124, USA; nicolette.prevost@usda.gov (N.T.P.); doug.hinchliffe@usda.gov (D.J.H.); sunghyun.nam@usda.gov (S.N.); sechin.chang@usda.gov (S.C.); rebecca.hron@usda.gov (R.J.H.); crista.madison@usda.gov (C.A.M.); jade.smith@usda.gov (J.N.S.); 2Department of Surgery, School of Medicine, Virginia Commonwealth University, Richmond, VA 23298, USA; poffenbergec@vcu.edu (C.N.P.); taylormm5@vcu.edu (M.M.T.); kirsty.dixon@vcuhealth.org (K.J.D.); 3Department of Internal Medicine, School of Medicine, Virginia Commonwealth University, Richmond, VA 23298, USA; erika.martin@vcuhealth.org

**Keywords:** wound dressings, cotton gauze, silver

## Abstract

The need for prehospital hemostatic dressings that exert an antibacterial effect is of interest for prolonged field care. Here, we consider a series of antibacterial and zeolite formulary treatment approaches applied to a cotton-based dressing. The design of the fabric formulations was based on the hemostatic dressing TACGauze with zeolite Y incorporated as a procoagulant with calcium and pectin to facilitate fiber adherence utilizing silver nanoparticles, and cellulose-crosslinked ascorbic acid to confer antibacterial activity. Infra-red spectra were employed to characterize the chemical modifications on the dressings. Contact angle measurements were employed to document the surface hydrophobicity of the cotton fabric which plays a role in the contact activation of the coagulation cascade. Ammonium Y zeolite-treated dressings initiated fibrin equal to the accepted standard hemorrhage control dressing and showed similar improvement with antibacterial finishes. The antibacterial activity of cotton-based technology utilizing both citrate-linked ascorbate-cellulose conjugate analogs and silver nanoparticle-embedded cotton fibers was observed against *Staphylococcus aureus* and *Klebsiella pneumoniae* at a level of 99.99 percent in the AATCC 100 assay. The hydrogen peroxide levels of the ascorbic acid-based fabrics, measured over a time period from zero up to forty-eight hours, were in line with the antibacterial activities.

## 1. Introduction

Hemorrhage is the leading cause of battlefield mortality and of civilian trauma age 1 to 46, and trauma-associated bacterial infections leading to sepsis are a major concern both on and off the battlefield [1,2]. Thus, the development of robust hemostatic dressings that also have antimicrobial activity is obviated in prehospital medicine both in civilian and military scenarios. This is especially the case with special forces operations that extend to remote and austere parts of the world where the evacuation of casualties is measured in days rather than hours [3]. Microbial growth on textiles that come in contact with the body may double at a rate of 20–30 min, causing undesirable effects and posing the potential for contamination to the user, especially when accessible medical care is inhibited. Moreover, the need for an effective hemostatic dressing that has robust antimicrobial activity is obviated by the virulent activity of *Staphylococcus aureus* which has evolved mechanisms to gain control over blood coagulation [4,5]. *S. aureus* is currently one of the deadliest infectious agents in the developed world, causing intravascular infections such as sepsis and infective endocarditis [6]. Thus, a robust, non-toxic antimicrobial dressing that is effective for prolonged care is ideally suited, and should: (1) accelerate clot formation and halt blood flow of hemorrhages in 2 min; (2) act as a barrier to microbial contamination and reduce bacterial colony formation; (3) be capable of remaining in place for 72–96 h without tissue breakdown, reducing the need for frequent dressing changes; (4) conserve tissue viability by providing a moist environment; and (5) prevent premature wound closure and the formation of fistulae.

Currently acute wound management in the prolonged field care setting is provided under guidance for the employment of hemostatic dressings which in some cases have shown some degree of antimicrobial activity [7]. For example, a number of chitosan and oxidized and regenerated cellulose materials that are listed as prehospital hemostatic dressings have been shown to provide mixed results, with some effective antibacterial activity promoted under the acidic conditions of the materials and fiber composition [8,9,10]. Although acute wounds tend to be slightly acidic, infected and chronic wounds have been found to increase in pH [11]. Given these issues, efforts to improve on chitosan formularies for combined hemostatic and antibacterial activity have been recently reported. For example, quaternary analogs of chitosan have been observed to promote antibacterial activity as a function of the degree of acetylation, molecular weight, and relative distribution of charge to the polymer backbone [12]. Dual hemostatic/antibacterial efficacy was reported in a mussel-inspired adhesion material containing gelatin and quaternized chitosan that was reported to promote good hemostasis while improving on antibacterial activity against *E. coli* and *S. aureus* [13].

Other recently reported dressing motifs have employed both synthetic and naturally occurring designs, employing nanotechnology and the use of sensors. A nanocomposite of carboxymethylated cellulose nanofibers treated with silver nanoparticles and gelatin was formed into a crosslinked hydrogel and reported to have good antibacterial and satisfactory hemostatic activity [14]. A cotton gauze dressing coated with polyvinyl alcohol nanofibers and loaded with ciprofloxacin demonstrated antibacterial activity against *E. coli* and *S. aureus*, with an improved blood coagulation index resulting from a longer acid-based oxidation of the cotton [15]. To improve both antibacterial and hemostatic activity while exerting an effect on the vascularization component of wound healing, a glucagon-like peptide analog was combined with zinc oxide on electrospun poly(L-lactide-co-glycolide) gelatin nanofiber. The authors report that the zinc oxide nanoparticles and glucagon analog act synergistically with the gelatin scaffold to improve hemostatic and antibacterial activity and progress in wound healing in a bacterial-infected wound model [16]. Notably, the incorporation of colorimetric sensors interfaced with antibacterial release systems has also been tethered to a hemostatic form of a copolymerized acrylic acid and porous starch polymer [17]. Kaolin, which is the active hemostatic agent in Combat Gauze, combined with poly(vinyl alcohol) sponges has been reported in adjuvant formularies combined with penicillin-streptomycin and marjoram [18].

As seen, these recent efforts to develop dual hemostatic and antibacterial dressing testify to research and development efforts to address the current gap where an effective technology is needed for prolonged field care. On the other hand, the mode of action of hemostatic procoagulant products and prototypes has been well characterized in recent years, and numerous procoagulants with hemorrhage control have been developed [13,19,20]. Some of the molecular-based forms of recombinant proteins are designed as mimics or derivatives and biosimilars of coagulation cascade factors including prothrombin and fibrinogen [20]. A systematic review of hemostatic agents for prehospital hemorrhage control recommended that Combat Gauze may be sufficient for controlling the majority of compressible traumatic wounds in non-coagulopathic patients, and fibrinogen-based dressings may offer the best option for the cessation of bleeding in coagulopathic patients [20,21]. However, the continued development of hemostatic materials that address the complex issues of trauma will require that material structure function relations governing binding with platelets and coagulation cascade proteins be optimized.

The underlying basic science that addresses the hemostatic process has advanced in recent years to yield mechanism-based approaches to enhance hemostatic dressing design. In the field of blood–surface interactions, the development of wound dressing materials has balanced efficacy with biocompatibility and toxicity. This paper addresses recent advances in cotton technology applied to complementing the procoagulant zeolite and auxiliary formulations that impart antibacterial activity. This is commensurate with a prolonged field care objective in so much as both hemostatic and antibacterial activity are required. Previously, we have addressed hemostatic material improvement based on procoagulant-treated fabrics that are blends of fibers designed by taking into consideration the material surface charge, polarity, gelation (swelling), and absorption capacity [22]. Moreover, we have previously shown the effectiveness of using both silver nanoparticles and ascorbic acid in greige cotton-containing fabrics to exert an antibacterial effect against both Gram-positive and -negative bacteria [23]. Taking these properties into consideration, we report here the hemostatic and antibacterial properties of cotton-based zeolite-containing dressings.

## 2. Results

We selected an ammonium zeolite formulary as the procoagulant and prepared it in tandem with two different antibacterial finishes that are based on in situ silver nanoparticle attachment and a chemistry that crosslinks ascorbic acid to cellulose. The impetus for this study was to examine the efficacy of the treated fabrics based on the bleeding control dressing TACGauze (TGz) treated with antibacterial designs and procoagulant zeolite formularies. Previously, we showed that ammonium zeolite demonstrated enhanced hemostatic efficacy [23,24]. Thus, we combined fabric treatment finishes employing two approaches that dial in hemostatic and antibacterial activity. Table 1 lists the fabric name and corresponding abbreviation and its treatment/method of modification. All fabrics employed were assembled by either commercially or pilot-based machinery at the USDA pilot facility at Southern Regional Research Center (SRRC) in New Orleans. The ratio of fibers of greige cotton/bleached cotton/polypropylene, 60/20/20, was blended and was subject to the hydroentanglement process as diagramed in the “Materials and Methods” section.

### 2.1. Antibacterial Activity

Two separate finishing formularies were employed to confer antibacterial properties to treat TGz, a nonwoven dressing for bleeding control. These employed traditional finishing chemistry treatments utilizing the pad-dry-cure methodology. Unbleached nonwoven cotton-based fabric finishes with free ascorbic acid and ascorbic acid attached through crosslinking to cellulose in the dressing fiber matrix have been previously shown to be effective against Gram-negative and -positive bacteria [25]. Silver nanoparticle-treated fabrics have also demonstrated antibacterial barrier properties in cotton-containing nonwovens, and were also employed in this study [26,27]. All of the antibacterial fabrics including those treated with zeolite, which was added to confer enhanced hemostatic activity, were effective at the 99.99 percent level against Gram-negative and -positive bacteria, as shown in Table 2.

**Table 1 ijms-24-17115-t001:** Description of fabrics.

Fabric Name	Abbreviation	Fabric Treatment
TACGauze	TGz	None
BIOGauze	BGz	Ascorbic acid on TACGauze applied using the pad-dry-cure method
Citrate TACGauze	CXTGz	Crosslinking citric acid and ascorbic acid to TACGauze using the pad-dry-cure method
Silver “TACGauze”	AgTGz	Silver “TACGauze” version manufactured at SRRC using cotton fibers embedded with silver nanoparticles ^1^
BIOGauze with zeolite	BGz +Y	Pad-dry application of ammonium Y zeolite (NH_4_Y) formulation to BIOGauze
Citrate TACGauze with zeolite	CXTGz +Y	Pad-dry application of NH_4_Y zeolite formulation to citrate gauze
Silver “TACGauze” with zeolite	AgTGz +Y	Pad-dry application of NH_4_Y zeolite formulation to AgTGz
TACGauze with zeolite	TGz +Y	Pad-dry application of NH_4_Y zeolite formulation to TACGauze

^1^ Silver (Ag) nanoparticles (NP) were introduced into cotton fibers using previously described methods by [26]. Silver-impregnated cotton fibers were substituted for the bleached cotton component (20%) of TACGauze, creating silver-impregnated TACGauze (AgTGz).

**Table 2 ijms-24-17115-t002:** Antimicrobial results of fabric samples tested according to the AATCC 100 test method ^1^.

Sample	*K. pneumoniae* (4352)	*S. aureus* (6538)
	% Reduction after 24 h	
TGz	0	0
BGz	>99.99	>99.99
CXTGz	>99.99	>99.99
AgTGz	>99.99	>99.99
BGz +Y	>99.99	>99.99
CXTGz +Y	>99.99	>99.99
AgTGz +Y	>99.99	>99.99

^1^ Conducted by Situ Biosciences.

### 2.2. Scanning Electron Microscopy (SEM)

SEM images for ascorbic acid, citrate, and AgTGz taken at lower magnifications reveal several commonalities between the samples. In these images, shown in Figure 1, cotton fibers are easily recognized by their signature ribbon-like morphology, while polypropylene fibers are identified by their smooth, cylindrical, and uniform appearance typical of extruded synthetic fibers. At a higher magnification of 1500×, the AgTGz sample demonstrates minor fibrillation compared to the ascorbic acid and citrate samples. This is most likely due to additional processing associated with the pretreatment of the cotton fibers to form silver nanoparticles. In the zeolite-treated samples, areas where zeolite has been both trapped between the fibers and adhered to the fiber surface are clearly visible. It is interesting to note that the ascorbic acid zeolite sample appears to contain more zeolite than the other zeolite-treated samples. This may be due to an ionic interaction between the ascorbic acid carboxylate and the ammonium ion of zeolite.

### 2.3. Fourier-Transform Infrared Spectroscopy (FT-IR)

The recorded FT-IR spectra of control TGz and treated samples, with and without zeolite, were analyzed and utilized the IR region displayed in Figure 2 and Figure 3. In Figure 2, wide bands ranging between 1650 and 1780 cm^−1^ appear for the fabrics treated in this study with the expected region of C=O from the COOH and C=C groups [28]. Based on theoretically predicted wavenumber, in the fabric treatment with ascorbic acid (BGz) and ascorbic acid with NH_4_Y zeolite (BGz +Y), a moderately intense band appears with a peak point at 1752 cm^−1^ and 1655 cm^−1^ for BGz and 1752 cm^−1^ and 1634 cm^−1^ for BGz +Y. The asymmetric and symmetric C=O stretching were assigned to the bands at 1752 cm^−1^, and the asymmetric and symmetric C=C stretching were seen as the bands at 1655 cm^−1^ and 1634 cm^−1^, respectively. The citrate gauze and its corresponding zeolite fabric, CXTGz and CXTGz +Y, showed increased intensity of the bands at 1707 cm^−1^ and 1721 cm^−1^, which was attributed to the absorbance of the carbonyl group (C=O) of citric acid.

In Figure 3, an intense peak at 999 cm^−1^ appears for the fabrics treated in this study with the expected region of the NH_4_Y zeolite formulation [24]. Remarkably, the spectra of BGz, CXTGz, and AgTGz (all red), and the fabrics treated with NH_4_Y zeolite such as BGz +Y, CXTGz +Y, and AgTGz +Y (all blue), contain an overlap in peak intensity absorbance with the TGz fabric (black line). The highest peak for the zeolite is a wide band centered at 999 cm^−1^. Glycosidic orientations in the cellulose and other polysaccharides present in the TGz gauze contain four overlapping peaks in the same range. As expected, the band near 999 cm^−1^ is observed as the most intense band in samples treated with the NH_4_Y zeolite.

### 2.4. Contact Angle

Figure 4 shows the water contact angles measured on the surface of various nonwoven fabrics. The average contact angle for TGz was 118.1°. In general, the surface is considered to be hydrophobic when the water contact angle is greater than 90°. The contact angle for AgTGz was 80.2°, which was smaller than that for TGz. AgTGz is composed of more hydrophobic fibers than TGz, i.e., 20% scoured and bleached cotton fibers used for TGz were replaced with silver nanoparticle-embedded raw cotton fibers for AgTGz. Considering that the content of silver nanoparticles is too small to affect the surface property of the fiber, the lower contact angle observed for AgTGz was associated with its web structure. AgTGz, which was fabricated at SRRC, has a larger porous structure. This is in line with the large standard deviation of contact angles measured from AgTGz. The zeolite treatment made all nonwoven fabrics immediately absorb a water droplet. On the other hand, the ascorbate treatment altered the surface of TGz to be hydrophilic, while the citrate treatment did not significantly modify the hydrophobic surface of the TGz.

### 2.5. Hydrogen Peroxide Generation

The finding that formulations on a greige cotton-based spunlaced nonwoven with ascorbic acid and silver produce antibacterial activity as shown in Table 2 is confirmed by the mode of action, as previously demonstrated with the generation of hydrogen peroxide from the ascorbic acid-treated fabrics (BGz and TGz, and their zeolite-treated analogs) in Figure 5. The graphs in Figure 5 present the hydrogen peroxide generation of the current studied fabrics from rapid onset to 48 h sustaining levels that are within antimicrobial efficacy (See Appendix A for actual values/determination for each fabric in graphic form, Appendix A). Thus, hydrogen peroxide is thought to be the mode of antibacterial action for the BGz and CXTGz and the analogous ammonium zeolite formularies [25]. On the other hand, the AgTGz, while demonstrating some hydrogen peroxide generation, is believed to have multiple modes of action such as the interaction of its embedded silver nanoparticles and their release to interact extra- and intra-cellularly [23,29,30]. In this regard, it is interesting that the levels of hydrogen peroxide produced per gram of fabric are consistent with the antibacterial results shown in Table 2, and to some extent with previous studies, in so much as the levels observed are associated with robust antimicrobial activity [25].

The two forms of antibacterial fabrics that produced relatively high levels of hydrogen peroxide generation are the fabrics treated with ascorbic acid alone (BGz) and citrate-crosslinked ascorbic acid, as shown in Figure 5. These gave a sustained release of hydrogen peroxide from 2.7 millimoles per gram of fabric to two logs less of hydrogen peroxide per gram of fabric for the duration of 48 h. Thus, there is a close similarity of hydrogen peroxide generation between the levels of ascorbic acid incorporated in the fabric finishes and the crosslinking of ascorbic acid through citrate to cellulose.

Although the addition of zeolite appeared to diminish the generation of hydrogen peroxide levels in the treated antibacterial fabrics, the lower levels observed for zeolite-treated fabrics were over one to two orders of magnitude, i.e., still within the limits of antimicrobial activity [31]. In addition, the silver nanoparticle cotton-blend fabrication produced, within 24 h, levels of hydrogen peroxide that exceeded the amount observed for untreated TGz. Even though lower levels of hydrogen peroxide are produced in zeolite’s presence, as shown in Figure 5, one can assume it did not impact the antibacterial activity negatively, as reflected in the results shown in Table 2.

### 2.6. Thromboelastography-Monitored Fibrin Formation and Absorption Capacity

Shown in Figure 6 are the graphs of the thromboelastography (TEG) results of time to fibrin formation (R values) for the treated fabrics to promote clotting using platelet-free human plasma (Stago, system control ‘N’, cat# 00678). The ammonium form of faujasite Y zeolite was selected as the procoagulant treatment because of previous results demonstrating its increased clotting efficacy of ammonium versus sodium counterion with bovine blood [24]. In Figure 6a, TGz fabrics finished with either sodium (NaY) or NH_4_Y zeolite, 5% or 10%, respectively, with the same amount of pectin and calcium chloride, 0.5% and 2%, respectively, along with application method, padded or sprayed, are compared for optimal formulation. These are tested with plasma against Combat Gauze (CG, Z-Medica, Wallingford, CT, USA) and TGz alone. The data show that Combat Gauze promoted the initiation of clotting, which was not significantly better than TGz treated with the ammonium form of zeolite Y, in combination with calcium chloride and pectin (ANOVA *p* < 0.0001); see Appendix A for statistical analysis, Appendix A. In Figure 6b, the antibacterial finishes and their zeolite analogs are compared to untreated TGz and control (no gauze, only plasma). It was observed that the antibacterial finishes alone (without ammonium zeolite Y) did not affect the ability of the plasma to undergo the initiation of clot formation, compared to untreated TGz. Conversely, the antibacterial finishes with ammonium zeolite Y did not impact the time of ammonium zeolite Y to initiate clotting compared to TGz alone (ANOVA *p* < 0.0001) (see Appendix A for multiple comparisons).

Related to the overall in vivo clotting performance of a dressing is its absorption capacity. However, this would not be expected to be reflected in the thromboelastography study of fibrin formation due to the absence of sufficient fabric volume. As shown in Table 3, the relatively high absorption capacity of TGz was retained in the crosslinked citrate-ascorbate-cellulose analog (CXTGz). This may be attributable to free unreacted citrate groups on the surface of the fabric promoting absorption analogous to carboxymethylcellulose [32]. The lower absorption capacity of BGz and AgTGz may be attributed to finishing steps and nonwoven fabric construction processes. Also notable was the reduction in absorption capacity observed for the zeolite-treated fabrics.

## 3. Discussion

The subject of this paper pertains to imparting to fabrics properties that accelerate hemostatic activity to enhance performance for severe bleeding control while exerting an effective and antibacterial effect. The design of the fabric modifications was based on combining antibacterial functionality with procoagulant efficacy. Though numerous studies have combined antibacterial and procoagulant functionality, reports are scarce regarding dressings that singularly and seamlessly halt severe hemorrhage in all coagulopathic conditions while effectively inhibiting bacterial growth while being kept in place for extended durations. Thus, our study focuses on gauze-type dressings containing an antibacterial finish with adherence of the procoagulant zeolite.

The dressing design outlined here may be preferred for bleeding from a narrow tract where maintaining placement for days may be required. Moreover, this type of dressing is easier to pack into the depths of the wound to make direct contact with the bleeding vessel. Although it has been noted that the field of hemorrhage control development has plateaued to some extent, there are still issues and concerns that are part and parcel of the complexities of trauma and safety related to thromboembolic complications, the transmission of blood-borne pathogens, and antibody response. Moreover, gauzes that provide high absorption capacity when applied through a ‘pool of blood’ are required for requisite field care of severe hemorrhage [33]. We believe that these are necessary aspects of trauma dressing technology that can be improved with gauze-based approaches for prehospital care.

The hemostatic profiles in this study were limited to a determination of fibrin formation. A comparison of the hemostatic activities of the materials revealed that the zeolite formularies produced similar results in all treated fabrics of this study. Analogously, the antibacterial treatments were all effective after 24 h (at the >99.99% level). On the other hand, among the untreated fabrics, the rate of fibrin formation was best for the CXTGz fabric where ascorbic acid was crosslinked with citrate. The time to fibrin formation (as shown in Figure 6b) was significantly shorter than the values determined for other untreated fabrics. It is interesting that CXTGz also had a relatively high contact angle, arguing for an improved initiation of fibrin formation. However, TGz did too. Potentially, the hydrogen peroxide generation of CXTGz could play a role in improved fibrin-based clot formation. Nonetheless, the effect of the changes in design created by the crosslinking of CXTGz will require examination for properties of hemostasis in future studies to help pinpoint the basis for the differences observed in this study.

Zeolite as a procoagulant is the active agent that increases and works synergistically with the fiber and fabric properties to enable clotting commensurate with severe hemorrhage control [24]. Zeolite has been shown to bind in the D domain of fibrinogen [34]. Thus, there is a mechanistic basis for combining formularies that contain procoagulants such as zeolite with dressing fabrics having fiber blends that provide more optimal surface polarity to initiate blood clotting.

Greige cotton-containing fabrics, such as the ones employed here, serve as a scaffold for zeolite while providing high absorption capacity. And these materials possess a hydrophobic surface, as demonstrated by the contact angle measurements of this study. A hydrophobic surface promotes hemostatic activity by increasing the rate of fibrin and clot formation. Increasing hydrophobicity in materials is associated with a shorter time to fibrin formation. In this regard, it is notable that materials with contact angle measurements from ninety degrees to one hundred and twenty degrees have a high degree of hydrophobicity. As demonstrated previously, fabrics having at least 30 percent of greige cotton and 20 percent of polypropylene give rise to contact angles that indicate hydrophobicity [35,36]. This is seen to some extent in this study, but in fabric finishes such as ascorbic acid, the nonwoven’s fibrous web porosity and the zeolite formulation application decreased the contact angle to below the threshold of hydrophobic character. Interestingly, higher hydrophobicity as measured by contact angle correlated with higher absorption capacity. It is important to note that fibrinogen, an abundant protein converted by thrombin to fibrin, promotes cell adhesion [37]. Fibrinogen also binds to hydrophobic surfaces [38]. However, hydrophilicity is also important for fluid acquisition, and bleached cotton is incorporated in the blends to enhance fluid uptake and absorbency. For example, it has been shown in studies conducted in a mixed hydrophilic and hydrophobic environment analogous to that present in the fabrics of this study, that contact activation initiates thrombin, and associated platelets that bind to hydrophobic surfaces give rise to a procoagulant effect [39]. This mechanism is synergistic with FactorXII (FXII)-initiated processes occurring on platelet surfaces [40].

Here, we employ thromboelastography to measure fibrin. Material properties such as charge, polarity, gelation (swelling), and absorption capacity influence the initiation of blood clotting [22]. Fibrin formation by way of FXII is thought to be the principal pathway to material surface contact activation. Positively charged amino acids in the heavy chain of the FXII protein bind to negatively charged surfaces. Upon binding to a negatively charged surface, the initiation of the conversion of FXII triggers serine proteases in the coagulation cascades [41]. FXII initiation also occurs on platelet surfaces [40]. Thus, there is a mechanistic basis for combining formularies that contain procoagulants such as zeolite with dressing fabrics having fiber blends that provide more optimal surface polarity to initiate blood clotting.

Clot formation is initiated by a synergistic process of fabric and fiber properties interacting with the molecular components of the coagulation cascade. In this regard, it is helpful to apply concepts on the mechanism of material-mediated blood contact activation to the formularies and fabric modifications of this study. The surface properties that promote blood clot formation include its hydrophobic and hydrophilic character, absorption capacity, gelation, and charge [39,42,43]. Thus, in these types of dressings, clotting is initiated on the fiber surface through contact activation of the coagulation cascade, and a mix of negative charge, hydrophobic, and hydrophilic character initiates material surface induced thrombosis [35,39,43,44]. It is also notable that contact activation by way of ‘adsorption-dilution’ may be evoked from competing proteins binding to polar surfaces [43]. Material swelling can also occur in a hydrated environment such as in blood, and is associated with gelling, as found in biopolymers [45]. For example, hemostatic biopolymers like chitosan and alginate give rise to gelation upon contact with blood, triggering hemostasis [46,47].

The need for early intervention for trauma-based infection in battlefield wounds has been well documented, based on trauma infectious disease outcomes reported from the war in Iraq and Afghanistan [48]. In a study that involved 2699 military hospital patients transferred from the battlefield, it was found that 34% developed a trauma-related infection during their initial hospitalization, with skin and soft tissue infections being the predominant forms. Of these, it was found that after discharge from U.S. hospitals, approximately one-third of the patients enrolled in the study developed a trauma-related infection within a median period of 88 days. Thus, trauma-related infections can often lead to complex wounds. Prognostic factors thought to be involved include the microbial load or bioburden, the specific types of bacteria in the wound, inter-bacteria interactions, and biofilm formation [49]. Moreover, biofilm formation is found in 60 percent of complex wounds [50,51]. Notably, hydrogen peroxide has been reported to be more effective against biofilm formation than quaternary ammonium compounds [52]. The antibacterial functionalities imparted in these studies have been introduced as integral to the use of greige cotton (unbleached cotton) nonwovens. Both silver nanoparticle and ascorbic acid attachment to cotton has been previously shown to work synergistically with the structure and composition of greige cotton to elicit robust antibacterial activity [25,26]. For example, silver nanoparticles are thought to act through multiple modes of action including damage to the cell wall, and intracellular mechanisms such as the inhibition of respiratory enzymes, dephosphorylation of tyrosine leading to inhibition of signal transduction, and soft acid/soft base mechanisms that effect phosphorous and sulfur in DNA [53]. Silver also initiates the release of ROS which may account for the levels of hydrogen peroxide generated in these studies. On the other hand, the antibacterial activity attributed to the ascorbic acid cotton-based hydrogen peroxide-generating fabrics of this study may be considered as an oxidative mechanism of action due to the generation of destructive free radicals. Hydrogen peroxide can undergo transport across bacterial cells and result in the scission of DNA [54]. However, the cell toxicity of hydrogen peroxide and the manner in which cells potentiate and inhibit its activity are a subject of current investigation [55].

## 4. Materials and Methods

### 4.1. Materials

CBV100 and CBV300 zeolites, purchased from Zeolyst International, Conshohocken, PA, USA, are the synthetic faujasite Y zeolite with the sodium cation (NaY) and the ammonium cation (NH_4_Y), respectively. The SiO_2_/Al_2_O_3_ molar ratios are 4.9–5.4 and 5.1 for NaY and NH_4_Y, respectively. The pectin (PEC), from citrus peel (≥74% galacturonic acid), l-ascorbic acid, and calcium chloride (CaCl_2_) were purchased from Sigma-Aldrich (now MilliporeSigma, Burlington, MA, USA). Citric acid anhydrous was purchased from Fisher Scientific. Sodium hypophosphite monohydrate (NaH_2_PO_2_·H_2_O) was purchased from JT Baker. All other chemicals and fabrics were from existing supply/inventory. Ultrapure water (18 Ω), Millipore, was used as solvent for formulations. The TACGauze (TGz) used was from H&H Medical, Williamsburg, VA, USA, a blend of 60% greige cotton/20% bleached cotton/20% polypropylene.

### 4.2. Production of Silver TACGauze (AgTGz) Fabric

#### 4.2.1. Synthesis of Silver Nanoparticle-Embedded Cotton Fibers

Silver nanoparticles were embedded into mechanically precleaned raw cotton fibers based on a previously published method [26,27]. Approximately 150 g of mechanically pre-cleaned raw cotton fibers was immersed in a 2 L aqueous solution of Triton X-100 (0.05 wt.%) for 10 min. The wetted fibers were centrifuged to remove the solution using a spin dryer. The fibers were then immersed in a 2 L aqueous solution of AgNO_3_ (0.4 mM) and heated at 100 °C for 1 h. The treated fibers were cooled down, centrifuged, and air-dried. The concentration of silver based on the dry cotton fibers was 480 ± 21 mg/kg.

#### 4.2.2. Nonwoven Fabric Production of Silver Nanoparticle-Embedded TGz

The staple fibers used to produce silver nanoparticle (AgNP)-embedded TGz prototype fabrics were mechanically cleaned greige cotton fibers commercially available as TruCotton (Wildwood Cotton Technologies, Greenwood, MS, USA) and polypropylene 1.5 Dpf × 38 mm (FiberVisions Inc., Duluth, GA, USA). The fibers were opened and blended by three passes through a feed hopper, axiflow cleaner, and Model 310 fine opener (Fiber Controls Corp., Gastonia, GA, USA). The fibers were blended at a ratio of 60:20:20 TruCotton, AgNP TruCotton, and polypropylene, respectively.

The blended fibers were then processed through a feed chute to form a batt which was fed into a 101.6 cm wide textile card fitted with Cardmaster plates (Saco-Lowell Corp., Boston, MA, USA). The fibrous web produced by the carding was then continuously fed through a crosslapper and needleloom (Technoplants srl., Pistoia, Italy). The crosslapper was set at 14 laps and the needleloom set at 135 points cm^−2^ and 500 strokes min^−1^ using 2 barb conical needles (Groz-Beckert KG, Albstadt, Germany), which produced needlepunched (NP) nonwoven fabric rolls ~1 m wide and ~65 g m^−2^.

The NP fabrics rolls were then fed through a 1 m wide hydroentanglement line consisting of an aquajet fitted with a mesh structuring drum (9.5 apertures per cm^−2^) and a gas fired single drum drying oven, as shown in Figure 7 (Trützschler Nonwovens GmbH, Dülmen, Germany). The aquajet utilized one low pressure jet head and two high pressure jet heads set at 3, 6, and 6 MPa, respectively. The jet strip used for the low pressure jet contained 40 orifices per inch with diameters of 0.12 mm, and the jet strips used for the two high pressure jets contained 50 orifices per inch with diameters of 0.12 mm. The processing speed was 5 m min^−1^. The hydroentangled fabric was fed through the drying oven at a temperature of 110 °C and then wound into rolls. At the specified processing parameters, the hydraulic energy exerted on the fabric was ~4 MJ kg^−1^, which was calculated as previously described [56].

### 4.3. Fabric Treatment Methods

#### 4.3.1. BIOGauze (BGz) and Citrate TACGauze (CXTGz) Production

The BGz formulation consists of 1% (*w*/*v*) l-ascorbic acid with 0.6% (*w*/*v*) 1-hexanol as a wetting agent in ultrapure water. The CXTGz formulation consists of 7% (*w/v*) citric acid, 5% (*w*/*v*) l-ascorbic acid, and 4% (*w*/*v*) sodium hypophosphite, with 0.6% (*w*/*v*) 1-hexanol as a wetting agent in ultrapure water. The TGz roll pks, gauze strips of 4.5 in w × 120 in l, were submersed and saturated in their respective formulations at a volume equaling 20× its weight. The strips were padded (Mathis) with kiss rollers at a pressure of 40 psi and speed of 1.4 m per minute to achieve a wet pick-up of approximately 120%. The strips were then dried using a continuous hot air dryer (<KTF>, Mathis) at 100 °C for 2 min. BGz was then cured at 160 °C for 1 min and CXTGz for 5 min. CXTGz had an additional step of rinsing the cured gauze in deionized water, padding to remove excess water, and drying at a temperature of 100 °C for 2 min.

#### 4.3.2. Zeolite-Treated Gauze

The respective gauze strips were submersed and saturated in a volume of the zeolite formulation 20× its weight. The zeolite formulation consisted of 0.5% (*w*/*v*) pectin (PEC), 2% (*w*/*v*) calcium chloride (CaCl_2_), and 10% (*w*/*v*) ammonium Y zeolite (NH_4_Y), with 0.6% (*w*/*v*) Triton-X100 as a wetting agent in ultrapure water. The strips were padded (Mathis) with kiss rollers at a pressure of 40 psi and speed of 1.4 m per minute to achieve a wet pick-up of approximately 120%. The strips were dried using a continuous hot air dryer (<KTF>, Mathis) at 120 °C for 3 min.

### 4.4. Hydrogen Peroxide Assays

The hydrogen peroxide levels generated by the fabrics were determined using the Invitrogen Amplex Red Hydrogen Peroxide/Peroxidase Assay Kit (Molecular Probes Inc., Eugene, OR, USA) as per the manufacturer’s instructions. Five mm diameter discs were prepared from each fabric type using a No. 149 Arch Punch (C.S. Osborne & Co., Harrison, NJ, USA). For each fabric type, enough discs to weigh ~10 mg were placed in a 2 mL microcentrifuge tube (4–5 discs), and a 20:1 dilution of phosphate-buffered saline (PBS: 0.1 M sodium phosphate, 0.15 M sodium chloride, pH 7.2) was added, e.g., 200 mL of PBS to 10 mg of fabric, for a final concentration of 50 mg mL^−1^. The tubes were incubated for 1, 5, 10, and 60 min, and 24 and 48 h with three replicates per fabric type and timepoint. A 20 mL aliquot from each tube was used for the assay, and the hydrogen peroxide concentration was determined using a Qubit 4 fluorometer (Thermo Fischer Scientific Inc., Waltham, MA, USA) at excitation and emission maxima of 571 nm and 585 nm, respectively. The negative controls for each timepoint consisted of 20 mL aliquots of PBS only with no fabric.

### 4.5. Scanning Electron Microscopy (SEM)

The fabric samples were affixed to aluminum stubs with carbon adhesive tabs and coated with gold to a thickness of 16 nm using a sputter coater (Luxor^AU^ Gold Sputter Coater, Aptco Technologies, Nazareth, Belgium). SEM images were taken using a desktop SEM unit (Phenom Pro G6, Thermo Fischer Scientific, Waltham, MA, USA) at magnifications of 500×, 1000×, and 1500×, with an accelerating voltage of 10 kV and pressure of 1.0 Pa.

### 4.6. Fourier-Transform Infrared Spectroscopy (FTIR)

The FTIR examinations of the untreated and treated gauze were performed using Agilent Cary 630 FTIR spectrometer (Agilent, Santa Clara, CA, USA) using the diamond attenuated total reflection (ATR) sampling accessory. The samples were placed on top of a type IIa diamond crystal (single reflection) and secured with the sample press. Each sample was examined at 3 different points chosen at random. A total of 32 scans were measured between 4000 and 650 cm^−1^, with a resolution of 16 cm^−1^ for each replicate. The spectra are presented as the average of three replicates, corrected for baseline, and normalized as stated in each figure. The spectral figures were prepared using OriginPro 2023b (OriginLab Corporation, Northampton, MA, USA).

### 4.7. Contact Angle

The contact angle of a water droplet on the nonwoven fabrics was measured using a contact angle analysis equipment (VCA Optima XE, AST Products, Billerica, MA, USA). A 1 μL drop of distilled water was syringed onto the fabric. The image of the drop was immediately captured and analyzed to yield a contact angle. The contact angles on ten different areas were measured and their average value was presented.

### 4.8. Thromboelastography

Thromboelastography (TEG) was performed using a TEG 5000 Thromboelastograph^®^ Hemostasis Analyzer System using the TEG analytical software 4.2.3 (Haemonetics Corporation, Niles, IL, USA). Samples of all materials were cut into 6 mm diameter discs using a biopsy punch (Miltex, Trenton, ON, Canada). The discs were then placed into a disposable cup in the TEG 5000 analyzer, and 20 µL of normal saline (0.9%) was added to “wet” the material. A volume of 340 µL of platelet free plasma (Diagnostica Stago, Inc., Parsippany, NJ, USA, system control ‘N’, cat# 00678) was combined with 20 µL of 200 mM CaCl_2_, and TEG analysis was performed at 37 °C. The TEG parameter measured in this study included the R-time (reaction-time), which is defined as the time required for initial fibrin formation. Multiple TEG runs were performed for each fabric material, and the data generated were analyzed using the Descriptive Statistics Data Analysis Tool of Microsoft^®^ Excel^®^ for Microsoft 365 MSO Version (2310) 32-bit.

### 4.9. Liquid Absorbent Capacity

The liquid absorbent capacity (LAC) of the samples was measured according to the EDANA/INDA test method NWSP 010.1.R0 (2015), using a nonwoven absorption tester Model GT-CN02 (Gester Instruments Co., Ltd., Dongguan, China). Five replicate measurements were performed for each fabric type and expressed as the mass of water absorbed as a percentage of the fabric mass.

### 4.10. Antibacterial Testing of Fabrics

The antibacterial activity of the finishes on the gauze strips was evaluated using a standard, globally recognized, antimicrobial textile test that provides a quantitative measurement degree of activity, namely, AATCC Test Method (TM) 100: Antibacterial Finishes on Textile Materials: Assessment of (American Association of Textile Chemists and Colorists, Research Triangle Park, NC, USA). AATCC TM 100 was performed by Situ Biosciences LLC (ISO 17025:2017(en) [57] and ISO 17034:2016(en) [58] accredited, Wheeling, IL, USA). Firstly, the cut-out disks of the treated nonwoven gauze strips were challenged with two method standard microorganisms, *Staphylococcus aureus* (4352) and *Klebsiella pneumoniae* (6538), separately, by incubating the microorganism inoculum in contact with the test sample for a duration of up to 24 h without drying. Following exposure, the inoculated microorganisms were recovered and the concentration of the organisms was determined, cfu/mL. The antimicrobial performance was determined by a comparison of the recovered organisms from the test and control samples at time 0 and at time 24 h, and was reported as a percent value relative to the control sample material, % Reduction, Log10 reduction. Situ Biosciences ISO 17034 Certified Reference Material (CRM) treated and untreated controls were used.

## 5. Conclusions

Prehospital dressings for battlefield and civilian trauma continue to be developed. Among recent new hemostatic dressings that have been reviewed, TGz has been cited as an affordable, viable alternative for severe hemorrhage in ‘worst case’ scenarios [19]. TGz is based on a light, highly absorbent nonwoven gauze that is composed of fibers of unbleached cotton with hydrophobic and hydrophilic fibers that combine to deliver a proprietary bleeding control technology. The work reported here demonstrates how the treatment of TGz with the procoagulant ammonium zeolite and the antibacterial functionality delivered through textile finishing chemistries result in an enhanced acceleration of clotting while exerting antimicrobial activity. The antibacterial activity is thought to proceed by way of two modes of action, the ascorbic acid generation of hydrogen peroxide and the silver nanoparticle inhibition of Gram-negative and Gram-positive bacteria. However, the actual mechanism of action requires more thorough examination. Notably, to establish the antibacterial and hemostatic efficacy of the dressing motifs of this study for prolonged field care, both an extended antibacterial dose response over 72 h and hemostatic in vivo assessments would be needed. Future work will also focus on testing for broader antimicrobial activity, as is indicated in the ESKAPE pathogen model. It is thought that the antimicrobial property of the dressings would allow the dressing to remain in place for extended periods of time with minimal risk to microbial contamination or damage to tissue. The manufacture of these types of dressings could be carried out in a straightforward manner using traditional textile finishing approaches. 

## Figures and Tables

**Figure 1 ijms-24-17115-f001:**
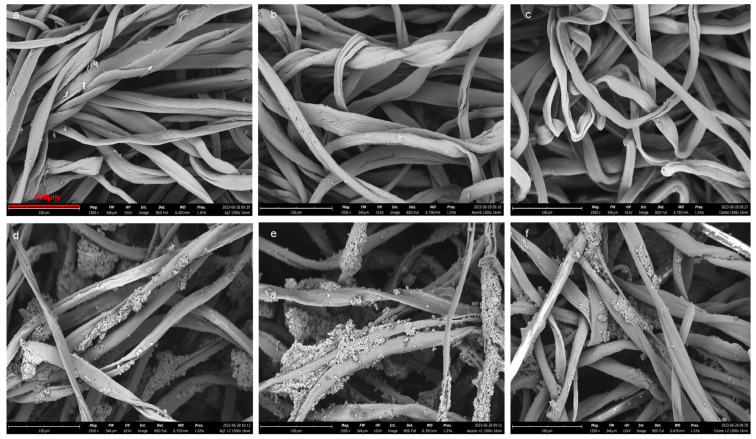
Scanning electron microscopy (SEM) micrographs of treated fabrics and their zeolite (NH_4_Y) analogs at 1500× magnifications: (**a**) AgTGz, (**b**) BGz, (**c**) CXTGz, (**d**) AgTGz +Y, (**e**) BGz +Y, and (**f**) CXTGz +Y.

**Figure 2 ijms-24-17115-f002:**
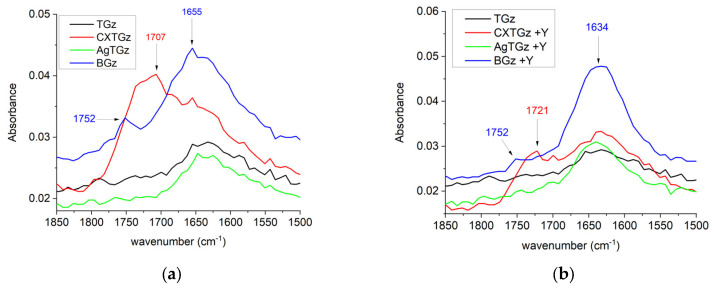
Fourier-transform infrared spectra (1850–1500 cm^−1^) of (**a**) control TACGauze (TGz, black), BIOGauze (BGz, blue), citrate TACGauze (CXTGz, red), and silver (AgTGz, green), and (**b**) control TACGauze (TGz, black), and the treated fabric with added zeolite formulation: BIOGauze with zeolite (BGz +Y, blue), citrate TACGauze with zeolite (CXTGz +Y, red), and silver with zeolite (AgTGz +Y, green).

**Figure 3 ijms-24-17115-f003:**
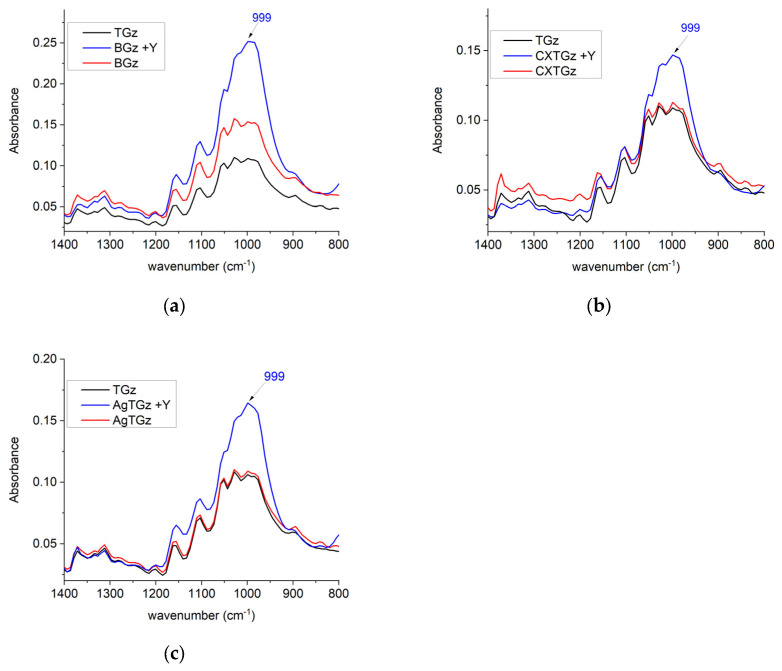
Fourier-transform infrared spectra (1400–800 cm^−1^) of untreated TGz, treated TGz strips, and its ammonium Y (NH_4_Y) zeolite analogs: (**a**) control (TGz, black), BIOGauze (BGz, red), and zeolite analog (BGz +Y, blue); (**b**) TGz (black), citrate TACGauze (CXTGz, red), and with zeolite (CXTGz +Y, blue); (**c**) TGz (black), silver TACGauze (AgTGz, red), and AgTGz with zeolite (AgTGz +Y, blue).

**Figure 4 ijms-24-17115-f004:**
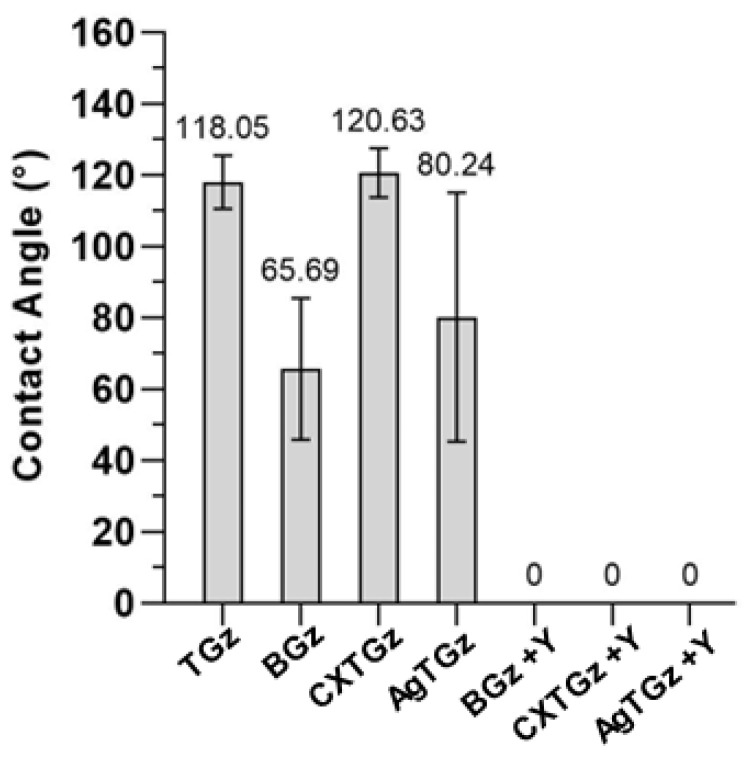
Contact angles of a water droplet on the surface of various treated TGz nonwoven fabric swatches. Fabric abbreviations are defined in Table 1.

**Figure 5 ijms-24-17115-f005:**
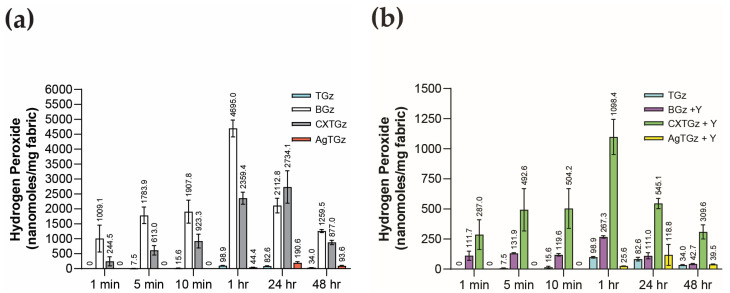
Quantitative determination of hydrogen peroxide generation of (**a**) TGz and its treated forms without the zeolite formulation and (**b**) with the zeolite formulation from time zero to 48 h. The graph bars of each selected timepoint are annotated with its mean concentration and standard deviation. Sample abbreviations are defined in Table 1.

**Figure 6 ijms-24-17115-f006:**
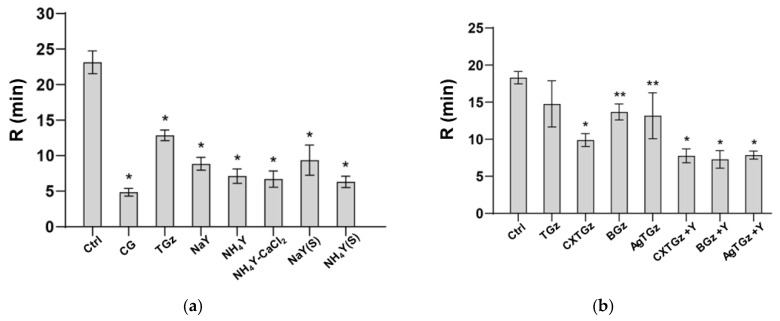
Both graphs present time to fibrin (R) formation of the TEG data of the treated TACGauze (TGz) fabrics compared to no gauze (ctrl), untreated TGz, and Combat Gauze (CG, Z-Medica). The bars represent the mean of each sample and are shown with standard deviations; the *p*-value is displayed with asterisks, * *p*-value < 0.001 and ** *p*-value < 0.05, as determined by one-way ANOVA with Dunnett multiple comparison test comparing the mean of each sample to the control mean in each group. (**a**) All zeolite formulations were applied to TGz swatches either by the padded or sprayed (S) method and all contain 0.5% pectin and 2% calcium chloride (CaCl_2_) with either 5% sodium Y (NaY) or 10% ammonium Y (NH_4_Y) zeolite. All samples were padded unless noted otherwise. (**b**) The description of these fabrics was stated in Table 1.

**Figure 7 ijms-24-17115-f007:**
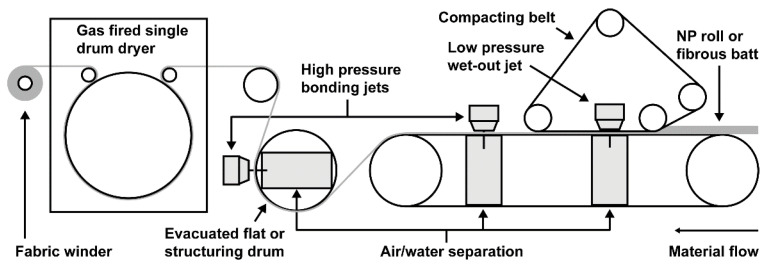
Schematic diagram of the Fleissner aquajet and drying oven used to produce hydroentangled nonwoven fabrics.

**Table 3 ijms-24-17115-t003:** Results of the liquid absorptive capacity testing of the treated TGz strips according to the EDANA/INDA test method NWSP 010.1.R0 (15).

Sample	Liquid Absorptive Capacity	
	%	SD ^1^
TGz	1016.40	13.37
BGz	887.59	76.66
CXTGz	1017.16	22.48
AgTGz	787.52	12.66
BGz +Y	610.40	25.62
CXTGz +Y	698.06	43.94
AgTGz +Y	522.63	24.06

^1^ SD: standard deviation.

## Data Availability

The data presented in this study are contained within the article and Appendix A.

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
