# Peer review of "Preparation and Activity of Hemostatic and Antibacterial Dressings with Greige Cotton/Zeolite Formularies Having Silver and Ascorbic Acid Finishes"

_ijms, 2023, doi:10.3390/ijms242317115_

Round 1
Reviewer 1 Report
Comments and Suggestions for Authors
The manuscript details the analysis of different fabrics designed to prevent both hemorrhages and bacterial infections in different clinical settings. The selected dressings were fully characterized for their structural properties but the microbiological part of the study is non-exixtent, both in materials and methods and results.
Even if the antibacterial tests were carried out by Situ Biosciences (as reported in Table 1), the methodology should be reported together with the information about the bacterial strains used in the study.
How can the Authors state that the antibacterial properties are "prolonged"? There are no results demostrating a time-correlation effect. In addition to the AATCC 100 test, is it possible to carry out antimicrobial investigations on discs prepared from the different fabrics (same procedure of the hydrogen peroxide assays) and analyse in more depth the inhibitory effect and the potency of the formulations? No differences by adding AgNPs or ascorbic acid?
In my opinion, the Authors should better emphasize their final results. They compared different dressings, what is the best for hemostatic properties? and for the antibacterial properties?
Author Response
"Please see the attachment"

Reviewer 2 Report
Comments and Suggestions for Authors
Comments to the Authors
Manuscript number Ijms-2629260-peer-review-v1.pdf “Preparation and Activity of Dressings Designed for Prolonged Field Care Use: Hemostatic and Antibacterial Properties of Greige Cotton/ Zeolite Formularies comparing Silver and Ascorbic Acid Finishes” presents a series of antibacterial and zeolite formulary treatment approaches applied to a cotton-based dressing. These fabric formulations were based on the hemostatic dressing TACGauze with zeolite Y incorporated as a procoagulant with calcium and pectin to facilitate fiber adherence utilizing silver nanoparticles, and cellulose crosslinked ascorbic acid to confer antibacterial activity. The FT-IR spectra were employed to characterize the chemical modifications on the dressing, the contact angle measurements to see the surface hydrophobicity on the cotton fabric. This hydrophobicity plays an essential role in contact activation of the coagulation cascade. Ammonium Y zeolite treated dressings initiated fibrin presented similar improvement with antibacterial finishes. The antibacterial activity of cotton-based technology utilizing both citrate-linked ascorbate-cellulose conjugate analogs and silver nanoparticle-embedded cotton fibers was determined against Staphylococcus aureus and Klebsiella pneumoniae at a level of 99.99 percent in the AATCC 100 assay. Hydrogen peroxide levels determined over a short-term of the ascorbic acid-based fabrics were in line with the antibacterial activities.
The manuscript should have some minor revision before publishing.
Here are some advices for the authors:
-In the Figures 2 and 3 (The FT-IR spectra may be it is better to remove the lines from the background.
-“The contact angle for AgTGz was 80.2°, which was smaller than that for TGz. AgTGz is composed of more hydrophobic fibers than TGz, ” can you explain better why the contact angles are different not how it was expected. If it’is more hydrophobic than it should be bigger.
-Figure 5…can you change the Y-scale for a better view of the hydrogen peroxide generation?
-There are some spelling mistakes.
-The references should respect the same format.
The study is interesting and if the manuscript would have some minor revision before publishing, it will be interesting for the readers of the International Journal of Molecular Sciences.
Author Response
"Please see attachment"

Round 2
Reviewer 1 Report
Comments and Suggestions for Authors
The scientific value of the manuscript would be improved by adding data on the prolonged antibacterial activity of the cotton/zeolite formularies, but, even in the present form, the paper has quality enough for publication.